TLR4 Asp299Gly (rs4986790) polymorphism and coronary artery disease: a meta-analysis

Chen Rui 1
Gu Ning 2 guning_2@163.com
Gao Ying 1
Cen Wei 1
1 The First Clinical College, Nanjing University of Chinese Medicine , Nanjing, Jiangsu , China
2 Department of Cardiology, The Third Affiliated Hospital of Nanjing University of Chinese Medicine , Nanjing, Jiangsu , China
Mehra Mandeep
Electronic publication date: 2015 Nov 26
Publication date: 2015
Volume: 3
Electronic Location ID: e1412
Received 2015 Aug 31; Accepted 2015 Oct 29
Copyright: © 2015 Chen et al.
Copyright year: 2015
Copyright holder: Chen et al.
License: This is an open access article distributed under the terms of the Creative Commons Attribution License, which permits unrestricted use, distribution, reproduction and adaptation in any medium and for any purpose provided that it is properly attributed. For attribution, the original author(s), title, publication source (PeerJ) and either DOI or URL of the article must be cited.
License URL: https://creativecommons.org/licenses/by/4.0/

Keywords: Coronary artery disease, Polymorphism, C-reactive protein, Statin, Stenosis, Toll-like receptor 4, Asp299Gly

Funding: National Natural Science Foundation of China 81173399 This work was supported by the National Natural Science Foundation of China (No. 81173399). The funders had no role in study design, data collection and analysis, decision to publish, or preparation of the manuscript.

==============================
Background. Previous studies have shown conflicting results on the association between toll-like receptor 4 (TLR4) Asp299Gly (rs4986790) polymorphism and coronary artery disease (CAD). The aim of this study was to evaluate the influence of TLR4 Asp299Gly polymorphism on CAD risk, CRP level and the number of stenotic coronary arteries, as well as to investigate whether G allele carriers would benefit more from statin treatment.

Methods. PubMed, EMBASE, and CNKI databases were searched until May 2015. All the statistical tests were performed using R version 3.1.2. Odds ratio (OR) and 95% confidence interval (CI) were used to assess the association between TLR4 Asp299Gly polymorphism and CAD risk, the number of stenotic vessels, and the incidence of cardiovascular events according to statin-treated patients. Weighted mean difference (WMD) was calculated for the association between Asp299Gly and CRP level.

Results. Overall, 12 case-control studies with 10,258 cases and 5,891 controls were included, and no association of TLR4Asp299Gly polymorphism with CAD was found (G allele vs. A allele: OR = 0.97, 95% CI [0.81–1.17], P = 0.75; AA vs. GG + AG: OR = 0.97, 95% CI [0.80–1.18], P = 0.76; GG vs. AG + AA: OR = 1.08, 95% CI [0.57–2.02], P = 0.82; AG vs. AA + GG: OR = 1.03, 95% CI [0.85–1.25], P = 0.74). Also, no association was noted between Asp299Gly and CRP level (WMD = −0.10, 95% CI [−0.62, 0.41], P = 0.69). Furthermore, no synergistic effect of statin and 299Gly was reported (Statin_AA vs. Statin_AG/GG: OR = 1.12, 95% CI [0.41–3.09], P = 0.82).

Discussion. This meta-analysis suggests no association of TLR4 Asp299Gly polymorphism with CAD and CRP level. It is further indicated that the G allele carriers may not benefit more from statin treatment. Further studies should include large sample size and high-quality literature to understand this issue in depth.

Introduction

Coronary artery disease (CAD), resulting from atherosclerosis (AS), has become the leading cause of disability and death globally (Murray et al., 2012). Evidence suggests that inflammation and immunity play a key role in the pathogenesis of AS and CAD (Ross, 1999; Libby, Lichtman & Hansson, 2013). Moreover, as a pattern recognition receptor of the innate immune system, toll-like receptor 4 (TLR4) expression is increased in human atherosclerotic lesions (Edfeldt et al., 2002). Various ligands (e.g., lipopolysaccharide (LPS), heat shock protein (HSP), minimally oxidized low-density lipoprotein (mmLDL), fibrinogen) can bind to TLR4, and then myeloid differentiation factor 88 (MyD88)-dependent or MyD88-independent signal pathway is activated, resulting in inappropriate immune activation, which consequently contributes to the onset and rupture of AS plaques (Dekker et al., 2010; Miller et al., 2012).

Human TLR4 is located in 9q32-q33 region, and contains three exons. TLR4 activity and function seem to be modulated by genetic variations, especially single nucleotide polymorphisms (SNPs) (Balistreri et al., 2009). Asp299Gly (+896A/G, rs4986790) is one of the only two SNPs in TLR4 that have a frequency greater than 5% in humans (Balistreri et al., 2009); the other is Thr399Ile (+1196C/T, rs4986791) and a high degree of linkage disequilibrium (LD) exists between them. As most of the variations, Asp299Gly is in the leucine-rich repeat (LRR) domain of Exon 3 which is associated with the recognition of pathogen-associated molecular patterns (PAMPs) (e.g., mmLDL) (Smirnova et al., 2000). Because of a glycine residue substituting for aspartic acid at amino acid position 299 (nucleotide substitution 896A > G), the extracellular domain of TLR4 is changed, leading to an attenuate signal pathway, blunted response to inhaled LPS, decreased production of inflammatory cytokines (Arbour et al., 2000), and reduced risk of the progression of atherosclerosis (Kiechl et al., 2002). Numerous studies have focused on the association between the Asp299Gly TLR4 polymorphism and CAD. Some reports suggested a protective effect of TLR4 Asp299Gly on CAD (Boekholdt et al., 2003; Ameziane et al., 2003; Kolek et al., 2004; Balistreri et al., 2004; Berg et al., 2009). In contrast, one study found that men with the 299Gly TLR4 genotype had an increased risk of myocardial infarction (MI) (Edfeldt et al., 2004), while other reports showed no obvious association between 299Gly and CAD (Morange et al., 2004; Zee et al., 2005; Koch et al., 2006; O’Halloran et al., 2006; Nebel et al., 2007; Beijk et al., 2010; Džumhur et al., 2012; Martínez-Ríos et al., 2013; Golovkin et al., 2014; Guven et al., 2015). Based on these, many studies have investigated the relationship between the severity of coronary artery stenosis and Asp299Gly, and obtained inconsistent results (Boekholdt et al., 2003; Yang, Holloway & Ye, 2003; Hernesniemia et al., 2006; Džumhur et al., 2012; Guven et al., 2015). Furthermore, based on the direct anti-inflammatory effect of statin (Crisby et al., 2001; Ridker et al., 2009) Boekholdt et al. (2003) stated that 299Gly can affect the efficacy of statin in preventing cardiovascular events, and thus the carriers of the variant allele benefit significantly from statin treatment; however, the same phenomenon was not found in other reports (Kolek et al., 2004; Beijk et al., 2010). The issue whether a synergistic effect of statin and 299Gly exists is still controversial. In addition, C-reactive protein (CRP) serves as a useful biomarker of inflammatory diseases, and is also considered as an independent risk factor to predict first and recurrent cardiovascular events; its level with TLR4 Asp299Gly has gained enormous attention (Kiechl et al., 2002; Edfeldt et al., 2004; Kolek et al., 2004; Netea et al., 2004; Hernesniemia et al., 2006; Beijk et al., 2010). In other words, a number of studies have been carried out on Asp299Gly and CAD, and the results are quite inconsistent. Thus, a meta-analysis is needed for further insights into the association between TLR4 Asp299Gly and the CAD risk, the CRP level and the number of stenotic coronary arteries; and to investigate whether a synergistic effect between statin and 299Gly exists.

Materials and Methods

Literature research

This meta-analysis followed the Preferred Reporting Items for Systematic Reviews and Meta-analyses (PRISMA) statement (Moher et al., 2010). PubMed, EMBASE, and CNKI databases were used to search relevant articles within a range of published years from January 1, 2000 to May 30, 2015. The full search strategy (for PubMed) about the association between TLR4 Asp299Gly polymorphism and CAD was as follows: “Toll-like receptor-4 or TLR4” AND “coronary heart disease or CHD or coronary artery disease or CAD or cardiovascular disease or CVD or myocardial infarction or MI” AND “polymorphism or variant.” The full search strategy about the association of Asp299Gly polymorphism with CRP level was as follows: “Toll-like receptor-4 or TLR4” AND “coronary heart disease or CHD or coronary artery disease or CAD or cardiovascular disease or CVD or myocardial infarction or MI” AND “polymorphism or variant” AND “CRP or C-reactive protein.” For the study of the synergistic effect between statin and 299Gly, the following search strategy was used: “Toll-like receptor-4 or TLR4” AND “coronary heart disease or CHD or coronary artery disease or CAD or cardiovascular disease or CVD or myocardial infarction or MI” AND “polymorphism or variant” AND “statin.” To avoid missing articles, the references cited in the research papers and review articles were examined as well.

Inclusion and exclusion criteria

The inclusion criteria for the studies about the association between TLR4 Asp299Gly polymorphism and CAD were as follows: (a) published case–control studies, and the control group should be the population without CAD; (b) clear diagnosis criteria of CAD; (c) studies supplied the number of individual genotypes in CAD cases and controls; (d) sufficient data for estimating an odds ratio (OR) or weighted mean difference (WMD) with 95% confidence interval (CI); (e) written in English or Chinese. The inclusion criteria for the studies about the association of TLR4 Asp299Gly polymorphism with CRP level, the number of stenosis coronary arteries, and the incidence of cardiovascular events with statin treatment were as follows: (a) integrated data; (b) sufficient data for estimating an odds ratio (OR) or weighted mean difference (WMD) with 95% confidence interval (CI); (c) measurement data had definite unit; (d) written in English or Chinese.

All reviews, case reports, animal studies and reports with incomplete data were excluded.

Data extraction and quality assessment

Data from the eligible studies were extracted by two authors based on the aforementioned criteria; if these two authors could not reach a consensus, the result was reviewed by a third author. Finally, the following information was recorded for each study: first author, year of publication, ethnicity, region, disease category, sample size, sex ratio, the number of allele and genotype counts of cases and controls, the frequencies of AA and AG/GG genotypes in patients with one, two, or three coronary arteries with >50% stenosis, the frequencies of AA and AG/GG genotypes in the patients with or without the incidence of cardiovascular events according to statin treatment, and the CRP level in AA and AG/GG groups. Whether these studies were in Hardy–Weinberg equilibrium (HWE) was also recorded (P < 0.05 was considered as a significant deviation from HWE). Quality assessment of studies included in the meta-analysis was conducted by two authors using the Newcastle–Ottawa Scale (NOS) (Wells et al., 2011). Scores were given for subject selection (i.e., adequateness of the case definition, representativeness of the cases, selection of controls, and definition of controls) and the comparability of the groups (i.e., comparability of cases and controls on the basis of the design or analysis) as well as measurement of exposure (i.e., ascertainment of exposure, same method of ascertainment for cases and controls, and non-response rate). NOS scores ranged from 0 to 9. Studies with a NOS score ≥ 6 were considered to be of high quality.

Statistical analysis

This meta-analysis used four gene models: allelic model (G allele vs. A allele), dominant model (AA vs. GG + AG), recessive model (GG vs. AG + AA) and super-dominant model (AG vs. AA+ GG), to explore the association between the CAD risk and TLR4 Asp299Gly polymorphism. In addition, OR and 95% CI were calculated to assess the association between Asp299Gly and CAD risk, the number of stenotic vessels, and the incidence of cardiovascular events according to statin treatment; WMD was calculated for the association between Asp299Gly and CRP level.

Heterogeneity among studies was evaluated using the Cochran’s Q statistic and the I2 statistic (P < 0.10 and I2 > 50% indicated evidence of heterogeneity) (Higgins & Thompson, 2002). If no heterogeneity in the data existed, the fixed-effects model was used; otherwise the random-effects model was used (Mantel & Haenszel, 1959; DerSimonian & Laird, 1986). Subgroup analysis (based on the type of CAD) and meta-regression were performed to explore the source of heterogeneity. In order to evaluate the stability of the results, sensitivity analysis was used, which meant omitting one study at a time, and then compared to show whether a significant difference existed between the former and the latter results. Furthermore, Begg’s funnel plot and Egger’s regression test were used to test publication bias (P < 0.05 was considered the representative of statistically significant publication bias) (Begg & Mazumdar, 1994; Egger et al., 1997). At last, cumulative meta-analysis was applied to reflect dynamic changes in the results according to different publication years. The statistical analyses were performed using metafor 1.9-5 (R 3.1.2).

Results

Association between Asp299Gly and CAD

Study characteristics

According to the PRISMA-statement flow diagram (Fig. 1), 17 full-text articles were assessed for eligibility, five (Boekholdt et al., 2003; Kolek et al., 2004; Holloway, Yang & Ye, 2005; Hernesniemia et al., 2006; Beijk et al., 2010) of which were excluded, because they were not proper case-control studies, despite being of high quality. Finally, 12 articles (NOS score > 6) (Ameziane et al., 2003; Balistreri et al., 2004; Edfeldt et al., 2004; Morange et al., 2004; Zee et al., 2005; Koch et al., 2006; O’Halloran et al., 2006; Nebel et al., 2007; Džumhur et al., 2012; Martínez-Ríos et al., 2013; Golovkin et al., 2014; Guven et al., 2015) were included with 10,258 cases and 5,891 controls. Most of the studies were conducted on Caucasian populations. The main characteristics of the studies are listed in Table 1.

Figure 1 Flow diagram of the study selection process.

Table 1 Characteristics of studies about the association between Asp299Gly and CAD.

Study	Year	Ethnicity	Region	Disease category	Sample size(ca/co)	Sex ratio (m/f)	Cases	Controls	HWE	
							AA	AG	GG	Sex ratio(m/f)	AA	AG	GG	Sex ratio(m/f)		
Martinez-Rios	2013	Mexican	Mexico	ACS	457/283	570/170	425	32	0	368/89	267	16	0	202/81	Y	
Ameziane	2003	Caucasian	France	ACS	183/216	353/46	169	14	0	160/23	187	28	1	193/23	Y	
				ACS			1,048	182	7							
O’Halloran	2006	Caucasian	Ireland		1,598/386	1,457/527				1,232/366	343	42	1	225/161	Y	
				SA			307	54	0							
Edfeldt	2004	Caucasian	Sweden	MI	1,164/1,508	1,838/834	1,038	126	0	821/343	1,374	133	1	1,017/491	Y	
Zee	2005	Caucasian	US	MI	370/695	1,065/0	323	46	1	370/0	605	87	3	695/0	Y	
Koch	2006	Caucasian	Germany	MI	3,657/1,211	3,385/1,483	3,283	360	14	2,772/885	1,069	138	4	613/598	Y	
Dzumhur	2012	Caucasian	Croatia	MI	119/120	171/68	104	15	0	81/38	98	22	0	90/30	Y	
Nebel	2007	Caucasian	Germany	MI	606/323	929/0	521	82	3	606/0	293	30	0	323/0	Y	
Balistreri	2004	Caucasian	Italy	MI	105/182	287/0	100	5	0	105/0	155	23	4	182/0	Y	
Morange	2004	Caucasian	France	CAD	247/490	737/0	211	35	1	247/0	439	50	1	490/0	Y	
Golovkin	2014	Caucasian	Russia	CAD	702/300	797/205	599	98	4	558/144	253	46	1	239/61	Y	
Guven	2015	Turks	Turkey	CAD	150/150	149/151	140	7	3	76/74	134	14	2	73/77	Y	
Notes.

ACS acute coronary syndrome

MI myocardial infarction

CAD coronary artery disease

HWE Hardy–Weinberg equilibrium

Y yes

N no

Sample size(ca/co) Sample size(cases/controls)

Sex ratio(m/f) Sex ratio(male/female)

Meta-analysis results of overall study

The heterogeneity of each gene model was as follows: allelic model (G allele vs. A allele): PQ = 0.0019, I2 = 61.34%; dominant model (AA vs. GG + AG): PQ = 0.0021, I2 = 61.13%; recessive model (GG vs. AG + AA): PQ = 0.97, I2 = 0.00%; super-dominant model (AG vs. AA + GG): PQ = 0.0039, I2 = 58.70%. Fixed-effects model was used for recessive model, and random-effects model was used for the other three models. G allele vs. A allele: OR =0.97, 95% CI [0.81–1.17], P = 0.75; AA vs. GG + AG: OR =0.97, 95% CI [0.80–1.18], P = 0.76; GG vs. AG + AA: OR =1.08, 95% CI [0.57–2.02], P = 0.82; AG vs. AA + GG: OR =1.03, 95% CI [0.85–1.25], P = 0.74. Overall, no association of TLR4 Asp299Gly polymorphism with CAD was observed. The results are shown in Fig. 2.

Figure 2 Forest plots of the association between TLR4 gene Asp299Gly polymorphism and CAD in different genetic models.

(A) allelic model: G allele vs. A allele; (B) dominant model: AA vs. GG + AG; (C) recessive model: GG vs. AG + AA; (D) super-dominant model: AG vs. AA + GG.

Heterogeneity analysis

The heterogeneity could not be effectively removed after subgroups were divided based on the different types of CAD. No association of Asp299Gly with CAD type was noted. To explore the source of heterogeneity further, meta-regression was used. It was found that ethnicity, region, disease category, sex ratio (male/female), and sample size ratio (cases/controls) were not the source of heterogeneity in all the models. In the allelic model, the frequency of A/G allele in the control group could explain 56.35% of I2, which meant part of the source of heterogeneity was explored. However, in the dominant model and the super-dominant model, the frequency of genotypes in the case group could explain 56.86% and 63.65% of I2 respectively.

Sensitivity analysis

One study was omitted at a time, and each of them made no obvious difference in the overall meta-analysis estimation, indicating that the results had a favorable stability.

Publication bias

Funnel plots (Fig. 3) intuitively reflected the publication bias. The results of Egger’s regression test were as follows: P = 0.25 for G allele vs. A allele, P = 0.25 for AA vs. GG + AG, P = 0.37 for GG vs. AG + AA, P = 0.16 for AG vs. AA + GG. Funnel plots with good symmetry and P value > 0.05 indicated no publication bias in the study.

Figure 3 Funnel plots of the association between TLR4 gene Asp299Gly polymorphism and CAD in different genetic models.

(A) allelic model: G allele vs. A allele; (B) dominant model: AA vs. GG + AG; (C) recessive model: GG vs. AG + AA; (D) super-dominant model: AG vs. AA + GG.

Cumulative meta-analysis

According to the order of published years, cumulative meta-analysis was performed, and it was found that OR value and 95% CI tended to be stable, and 95% CI gradually narrowed, as shown in Fig. 4.

Figure 4 Cumulative meta-analysis of the association between TLR4 gene Asp299Gly polymorphism and CAD in different genetic models.

(A) allelic model: G allele vs. A allele; (B) dominant model: AA vs. GG + AG; (C) recessive model: GG vs. AG + AA; (D) super-dominant model: AG vs. AA + GG.

Association between Asp299Gly and the number of stenotic coronary arteries

No association was found between the genotype frequency and the number of coronary arteries with >50% stenosis. Details are shown in Article S1.

Association between Asp299Gly and CRP level

Study characteristics

Six reports (Kiechl et al., 2002; Edfeldt et al., 2004; Kolek et al., 2004; Netea et al., 2004; Hernesniemia et al., 2006; Beijk et al., 2010) were included, in which the CRP levels of different genotypes (AA, AG/GG) were recorded using median and interquartile ranges or x¯±s. According to the method provided by an article (Hozo, Djulbegovic & Hozo, 2005), median and interquartile ranges were converted into x¯±s for the convenience of statistics. Details are shown in Table 2.

Table 2 Characteristics of studies about the association between Asp299Gly and CRP level.

Study	AA(mg/l)	GG/GA(mg/l)	
	Mean	SD	N	Mean	SD	N	
Kolek, 2004 (mg/dl)	1.23	0.913	1,725	1.11	0.873	169	
Edfeldt, 2004	1.5	1.67	1,791	1.6	2	186	
Beijk, 2010 (GENDER)	5.8	0.34	2,344	6.6	0.81	338	
Beijk, 2010 (GEISHA)	5	0.5	202	5.5	1.4	22	
Hernesniemi, 2008	1.92	4	1,812	1.6952	2.7685	389	
Netea, 2004	3.8	5.6	261	6.5	10.6	32	
Kiechl, 2002	3.72	7.8	755	2.4455	3.7465	55	

Meta-analysis results

Test for heterogeneity: PQ < 0.0001, I2 = 96.20%, random-effects model was used. WMD = −0.10, 95% CI [−0.62–0.41]. P = 0.69. Egger test: P = 0.77.

Also, no significant difference was found between Asp299Gly and CRP level. Forest plot and funnel plot are shown in Fig. 5. Larger heterogeneity might be caused by different measurement methods for CRP. Due to the lack of raw data, we couldn’t know all measurement methods of CRP, namely, we couldn’t remove the heterogeneity, so the persuasion of this result was limited.

Figure 5 Forest and Funnel plots of the association between TLR4 gene Asp299Gly polymorphism and CRP level.

(A) Forest plot; (B) Funnel plot.

Synergistic effect of statin and Asp299Gly

No synergistic effect of statin and 299Gly was found. Details are shown in Article S2.

Discussion

The worldwide distribution of G allele is closely related to the ethnic groups, which might be the result of differences in environmental pressure during human migration (Ferwerda et al., 2007; Ferwerda et al., 2008). TLR4 Asp299Gly polymorphism is rather rare in Asian populations (Lin et al., 2005; Nakada et al., 2005; Kim et al., 2008; Yuan et al., 2010), while 6%–14% of the Caucasian population is positive for Asp299Gly (Balistreri et al., 2009). For this reason, previous studies were mainly carried out in Europe. To clarify the relationship between CAD and Asp299Gly more effectively, it is worth mentioning that in the present meta-analysis, reports about Asp299Gly from Mexican and Turkish populations were included for more reliable studies. This meta-analysis showed no association between Asp299Gly and CAD. Subsequent subgroup analysis and sensitivity analysis, as well as the study on the association between Asp299Gly and the number of stenotic coronary arteries all confirmed this conclusion. This result was also consistent with the conclusions of the other two previous meta-analyses (Zhang et al., 2012; Yin et al., 2014).

In addition, CRP, a nonspecific acute phase protein, is a useful biomarker of inflammation. It is also considered as an independent risk factor to predict first and recurrent cardiovascular events such as MI or stroke (Kaptoge et al., 2012; Wennberg et al., 2012). Furthermore, in a randomized trial, statin therapy significantly reduced the incidence of major cardiovascular events among people without hyperlipidemia but with elevated CRP level (Ridker et al., 2008), which further supported the anti-inflammatory effect of statin. Thus, the present meta-analysis analyzed the association of Asp299Gly with CRP level, and explored whether G allele carriers could benefit more from statin treatment. It showed no obvious association between Asp299Gly and CRP level, and G allele carriers could not benefit more from statin treatment. However, the conclusion was underpowered due to the insufficient number of articles included and large heterogeneity of this meta-analysis. Hence, future studies should include large sample size and high quality literature to understand this issue in depth.

This meta-analysis has certain limitations as follows: first, CAD is influenced by multiple genetic mutations (Enquobahrie et al., 2008), so some interaction among genetic variations might exist widely. For example, some articles (Morange et al., 2004; Vainas et al., 2006) considered the synergistic effect of TLR4/Asp299Gly and CD14/C-260T, and one of them (Vainas et al., 2006) found that the carriers with TLR4 G allele/CD14 TT genotype, rather than each SNP individually, were associated with the atherosclerotic disease. Also, some reports (Boekholdt et al., 2003; Edfeldt et al., 2004; Koch et al., 2006; Guven et al., 2015) were available about the combination effect of TLR4/Asp299Gly and TLR4/Thr399Ile. So, it can be speculated that the interaction among SNPs in TLR4 and TLRs and other signal molecules may exist. Some literature suggests that the analysis of a number of polymorphic genetic markers is more informative than the analysis of a single polymorphism (Olivieri et al., 2006); however, this meta-analysis did not involve the interaction among SNPs. Second, CAD is a multifactorial disease, and its incidence and development are closely related to environmental and life-style factors. Hence, a small contribution of a SNP to CAD might be obscured by the presence of various dominant risk factors (Incalcaterra et al., 2013). Patients with CAD with different gender, age, and life-style have different pathophysiological characteristics (Edfeldt et al., 2004; Olivieri et al., 2006; Incalcaterra et al., 2010). Some articles have considered the influence of these factors, while othershave not. Due to the lack of original data, the influence of life-style (e.g., smoking) on the results of the meta-analysis was not investigated. Third, although the meta-analysis was based on detailed inclusion and exclusion criteria, some important uncontrollable factors still existed, such as different study design, different source of controls, and different environment. This might be the reason why the heterogeneity could not be removed effectively after subgroups were divided based on different types of CAD.

In conclusion, this meta-analysis suggested no association of TLR4 Asp299Gly polymorphism with CAD and CRP level. Moreover, the findings indicated that the G allele carriers might not benefit more from statin treatment. However, these results should be confirmed by conducting more high-quality studies in future.

Supplemental Information

Article S1 Association between Asp299Gly and the number of stenotic coronary arteries

Click here for additional data file.

Article S2 Synergistic effect of statin and Asp299Gly

Click here for additional data file.

Supplemental Information 1 PRISMA2009Checklist.doc

Click here for additional data file.

Supplemental Information 2 R code and related raw data

Click here for additional data file.

We thank Zhibin Quan (School of Computer Science and Engineering, Southeast University) for technical discussion.

Additional Information and Declarations

Competing Interests

Author Contributions

Data Availability

The authors declare there are no competing interests.

Rui Chen conceived and designed the experiments, performed the experiments, analyzed the data, contributed reagents/materials/analysis tools, wrote the paper, prepared figures and/or tables, reviewed drafts of the paper.

Ning Gu conceived and designed the experiments, performed the experiments, reviewed drafts of the paper.

Ying Gao analyzed the data, reviewed drafts of the paper.

Wei Cen wrote the paper, reviewed drafts of the paper.

The following information was supplied regarding data availability:

The research in this article did not generate any raw data.

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
