# Peer review of "TLR4 Asp299Gly (rs4986790) polymorphism and coronary artery disease: a meta-analysis"

_PeerJ, doi:10.7717/peerj.1412_

## Round 0.1 · original submission · Minor Revisions

· Academic Editor

Minor Revisions

The report requires close attention to editing by an expert English native speaker. Also the additional analysis breakdown proposed by a reviewer should be performed. Areas where the data is acquired from cohorts without nested power should be stated as such.

Reviewer 1 ·

Basic reporting

The paper is present well with good structure.

Experimental design

Well performed meta-analysis following mostly the protocol of most previous meta-analyses.

Validity of the findings

The findings have very little novelty value but settling the matter of false positive reporting on the polymorphims of TLR4 gene associating with CAD is commendable.

Additional comments

The meta-analysis by Chen et al. deals with the possible effects of TLR4 gene polymorphism. The main result is, that while few studies published ten years ago indicated that Asp299Gly polymorphism could be associated with the risk of CAD, a meta-analysis of all studies conducted since then shows no significant association. Publishing negative results is commendable although it is has already become evident on a general level that most results of genetic association studies are not replicable.


I have very little to add to this report. It is mostly well written but could use some revision for improving the language.

Please present present currently used rs-number of the Asp299Gly polymorphism in the Title and Abstract.

If possible, some of the result could be presented in supplementary material. For example the part of possible modulating effect of statin treatment could be transferred to supplements as well as the section dealing with “Association between Asp299Gly and the number of stenosed coronary arteries”.

Reviewer 2 ·

Basic reporting

Meta-analysis is correctly conducted and litterature is consistent. Minor revisions are required :
-Language: grammar and spelling should be revised
-It should better compare : 1-vessel+2-vessel vs. 3-vessel , it may give more significant result than (1-vessel+2-vessel+3-vessel vs. 3-vessel.
-Conclusion about incidence of cardiovascular events according to TLR4 genotype and statin treatment is underpowered because only three studies were included (indicate in discussion)

Experimental design

Meta-analysis correctly conducted

Validity of the findings

Although results are negative, they are valid.

Additional comments

no comment

---

## Round 0.2 · accepted · Accept

· Academic Editor

Accept

What we meant by "nested power" was to include a comment regarding whether the individual studies included in the analysis themselves had sufficient power to answer the questions posed. You may wish to add a statement to that effect in the galley state but I do not feel strongly about this.